# Simulation and Mechanical Properties of Fine-Grained Heat-Affected Zone Microstructure in 18CrNiMo7-6 Steel

**DOI:** 10.3390/ma15196782

**Published:** 2022-09-30

**Authors:** Tomaž Vuherer, Fidan Smaili, Edvard Bjelajac, Mirza Manjgo, Gorazd Lojen

**Affiliations:** Faculty of Mechanical Engineering, University of Maribor, Smetanova ulica 17, 2000 Maribor, Slovenia

**Keywords:** weld joint, fine-grained HAZ, simulation of microstructure, hardness, impact toughness, tensile properties, fatigue crack growth, 18CrNiMo7-6 steel

## Abstract

Heat-affected zones (HAZs) in real welds are usually quite narrow, and consequently most standard mechanical tests are difficult or even impossible. Therefore, simulated microstructures are often used for mechanical tests. However, the most often used weld thermal cycle simulator produces only a few millimeters wide area of simulated microstructure in the middle of specimens. Consequently, these kind of simulated specimen are not suitable for standard tensile tests, and even for Charpy impact tests, the simulated area can be too narrow. Therefore, to investigate the mechanical properties of a fine-grain heat-affected zone in 18CrNiMo7-6 steel, two methods were used for simulation of as-welded microstructures: (a) a weld thermal cycle simulator, and (b) as an alternative, though not yet verified option, austenitizing in a laboratory furnace + water quenching. The microstructures were compared and mechanical properties investigated. The grain sizes of the simulated specimens were 10.9 μm (water-quenched) and 12.6 μm (simulator), whereby the deviations from the real weld were less than 10%. Both types of simulated specimen were used for hardness measurement, Charpy impact tests, and fatigue tests. Water-quenched specimens were large enough to enable standard tensile testing. A hardness of 425 HV, yield strength *R*_p02_ = 1121 MPa, tensile strength *R*_m_ = 1475 MPa, impact energy *KV* = 73.11 J, and crack propagation threshold Δ*K*_thR_ = 4.33 MPa m^0.5^ were obtained with the water quenched specimens, and 419 HV, *KV* = 101.49 J, and Δ*K*_thR_ = 3.4 MPa m^0.5^ with the specimens prepared with the simulator. Comparison of the results confirmed that the annealed and quenched specimens were suitable for mechanical tests of FG HAZs, even for standard tensile tests. Due to the use of simulated test specimens, the mechanical properties determined can be linked to the FG HAZ microstructure in 18CrNiMo7-6 steel.

## 1. Introduction

Since welding is one of the most used joining processes, good understanding of the factors that affect the quality of welded joints is critical. Due to the heat generated by the arc between the electrode and the base material, changes occur in the microstructure of the base material that can manifest as modification of grain size, dissolution and/or precipitation of secondary phases, inhomogeneity, occurrence of cracks, and other types of defects, whereby the reliability and load-carrying capacity of the welded component decreases. As the present defects might not be detectable by nondestructive inspection methods, the mechanical properties of weld joints must be verified by tensile tests, hardness measurements, and impact toughness tests, and in the case of dynamic loads, also with fatigue tests.

Improvements in properties and the study of crack behavior of dynamically loaded materials and weld joints in a variety of steel types has attracted the attention of numerous authors [1,2,3,4,5,6,7,8,9,10,11,12,13,14,15]. In previous research [16] on martensitic steel P91, it was found that in the as-welded condition, the IC HAZ (intercritical HAZ) exhibited significantly higher impact toughness than the FG HAZ (fine-grained HAZ), while in the CG HAZ (coarse-grained HAZ), it was the lowest. The 18CrNiMo7-6 steel has attracted attention as well [17,18,19,20,21]. Although 18CrNiMo7-6 steel is used frequently, not much research has been dedicated to the mechanical properties of welds in this steel, perhaps because it is used predominantly for highly strained gear parts. Vuherer [18] and Vuherer et al. [22] studied the properties of a coarse-grained HAZ (CG HAZ) in 18CrNiMo7-6 steel with rotary bending fatigue tests. Smaili et al. [23] also researched the mechanical properties of the CG HAZ of this steel, but they used notched and precracked specimens according to ASTM E 647. Much fewer data are available on FG HAZs in 18CrNiMo7-6 steel. An FG HAZ was part of the investigation in [23]; however, only one type of simulated specimen was used. Therefore, this manuscript is focused only on FG HAZs.

One of the factors that influence the mechanical properties is microstructure. It is crucial that the whole HAZ and weld metal have adequate properties to resist external loads [23]. The only way to identify the weakest area and to take measures to improve its properties is to investigate the weld metal and each HAZ subzone separately.

Unfortunately, the width of individual HAZ subzones is very small. Even the total width of an HAZ can be only a few tenths of a millimeter [24] if welded with low heat input. Therefore, apart from microhardness measurements, mechanical testing of real-weld HAZ subzones is unreliable, or even impossible [12,24,25]. When using test pieces cut from a real-weld HAZ for destructive testing, it is practically impossible to assure a homogeneous microstructure in the whole area of rupture, and it is also practically impossible to manufacture two test pieces with identical microstructures in the area of rupture. Furthermore, even microstructures in the vicinity of the fracture surface exhibit an influence. Consequently, the scattering of results is enormous, and the results cannot be linked to only one certain type of microstructure. To obtain reliable data on mechanical properties in relation to microstructure, a larger volume of homogeneous microstructure is inevitable. The most suitable way to produce sufficiently large volumes of material exhibiting a homogeneous microstructure of a certain type is the simulation of microstructures [12,24,25,26]. Several research groups compared the microstructures and hardness of real-weld HAZs with simulated HAZ microstructures and confirmed the suitability of the simulated material for mechanical tests [12,24,27]. Good agreement on microstructures and hardness of the simulated HAZ subzones with real HAZ subzones was established when the real workpiece and the simulated material experienced equal thermal cycles. Considering all this, it was decided to use simulated microstructures in the present research.

Generally recognized as appropriate and used most widely is a weld thermal cycle simulator (WTCS). Unfortunately, in WTCS specimens, only a narrow section in the middle of a specimen exhibits the simulated microstructure. Consequently, WTCS specimens can be recommended without limitations only for investigation of microstructure, hardness measurements, and fatigue tests, where the plastic zone around the crack tip is very small. They are not suitable for standard tensile tests. Even the results of Charpy impact tests obtained with WTCS specimens can be unreliable, because the plastic zone around the crack tip in ductile materials can be quite wide, wider than the simulated area. The influence of this phenomenon is the strongest in ductile materials if the simulated area exhibits higher hardness and strength than adjacent areas, such as in the case of simulated FG HAZs. In such cases, the plastic deformation takes place partially in the softer, more ductile surrounding material, and a certain fraction of the impact energy is absorbed outside the simulated area. This reflects notably in higher measured total impact energy and even more strongly in higher propagation energy. Therefore, an alternative method that can produce significantly larger volumes of simulated material would be welcome.

As is well known for annealing treatments and tempering, the same effect can be achieved with different combinations of time and temperature only if the value of the tempering parameter, e.g., Hollomon–Jaffe parameter (HJP) [28] (p. 245), remains the same. Karmakar and Chakrabarti [29], Euser et al. [30], and Judge et al. [31] used the HJP for comparison of properties of conventionally annealed steel with a rapid annealed steel. In those works, hating rates were up to 1000 °C/s and soaking times from 1 s upwards. At equal HJP, the hardness of the samples was always comparable; however, the coincidence of tensile strength, impact toughness, and microstructures was not always that good. This means that if for simulations of HAZ microstructures, alternative techniques were to be be applied, not only HJP and hardness should be comparable to that obtained with the WTCS but also the microstructures must be comparable. Such an approach was already used successfully for simulation of CG HAZs [18]. In that work, austenitizing in a laboratory furnace (AF) + water quenching (WQ) was used to obtain a CG HAZ microstructure. The annealing temperature was significantly lower than the peak temperature in the real-weld CG HAZ, only 1100 °C, and the soaking time was 3 h, to obtain the same austenite grain size as in the real-weld CG HAZ. The temperature of samples during WQ was measured continuously with thermocouples welded on the surface of the test pieces, and a cooling rate was equal to that measured in the real-weld CG HAZ. The hardness, grain size, and dislocation density in the real-weld CG HAZ were compared with those in the simulated samples and a good match of all properties was observed.

Therefore, in the present research, two methods for simulation of as-welded FG HAZ microstructures were applied: the usual WTCS, and as an alternative, austenitizing in a furnace + water quenching (AF + WQ), which enables preparation of specimens long enough for standard tensile tests. For verification of the alternative AF + WQ method, the results obtained with the AF + WQ specimens were compared with the results obtained with the well-established WTCS method and with a real-weld HAZ.

The investigation was planned as follows. Suitable welding parameters were determined for real welding with a heat input as low as possible. Then, real welding was performed in order to produce reference microstructure and to record the thermal cycle. Consequently, the WTCS specimens were made with a weld thermal cycle simulator. A suitable combination of annealing temperature and duration to assure adequate grain size in the AF + WQ specimens was determined by tests. Adequate quenching parameters to ensure the desired cooling rate were also determined by tests. Preparation of the annealed and quenched specimens followed. After microscopy and mechanical tests, the results obtained with both types of simulated specimens were compared to confirm (or refute) the suitability of the AF + WQ specimens for characterization of the mechanical properties of an FG HAZ.

## 2. Materials and Methods

### 2.1. Materials

The chemical composition of 18CrNiMo7-6 steel, as stated in the material certificate, is given in Table 1. A continuous cooling transformation diagram (CCT diagram) of 18CrNiMo7-6 steel [32] is shown in Figure 1. Temperature Ac1 was 705 °C and temperature Ac3 was 790 °C. In other sources, e.g., [33,34], very similar CCT diagrams can be found for this steel.

### 2.2. Experimental Procedure

According to ISO/TR 15608-1 [35], 18CrNiMo7-6 steel belongs to group 3. According to EN ISO 15614-1 [36], for group 3 the admissible maximum hardness in as-welded condition is 450 HV. Hence, according to the CCT diagram, the minimum admissible Δ*t*_8/5_ is around 10 s. To limit the coarsening of the microstructure, it was decided to work with the lowest-possible heat input. Consequently, Δ*t*_8/5_ = 10 s was selected, and other parameters were calculated according to SIST EN 1011-2, Annex D6 [37], for two-dimensional heat flow. A single-V butt weld was made on a 15 mm thick preheated plate to produce a reference microstructure. During welding, the cooling time Δ*t*_8/5_ was measured with preinstalled thermocouples. The parameters are summarized in Table 2.

Two groups of specimens with fine-grained microstructures were prepared, artificially simulated using (a) a weld thermal cycle simulator (WTCS) and (b) austenitizing in a laboratory furnace followed by water quenching (AF + WQ). The weld thermal cycle simulator Smitweld 1405 and laboratory furnace Bosio EUP-K 20/1200 were used, respectively. The goal was to achieve the same Δ*t*_8/5_ and the same martensitic FG HAZ microstructure as observed in a real-weld FG HAZ, where the average grain size is 11.5 μm. The thermal cycles of the simulated FG HAZ specimens are presented graphically in Figure 2, while the parameters are summarized in Table 3.

For the AF + WQ specimens, a peak temperature lower than for WTCS specimens was selected to obtain equal grain sizes in both types of specimens, but still sufficiently above A_C3_ to assure homogeneous austenite. In a furnace, specimens are inevitably exposed to high temperatures longer than in a real weld or in the thermal cycle simulator because they heat up more slowly, and a certain soaking time is necessary to ensure a homogeneous microstructure across the whole cross section. Consequently, the temperature had to be lower than in real welding to prevent excessive grain growth. With preliminary experiments, it was established that the same austenite grain size as in the real-weld thermal cycle can be obtained in 15 min (5 min for heating + 10 min soaking time) in a furnace preheated to 870 °C.

In both WTCS and AF + WQ, the cooling rates were controlled via thermocouples welded onto the specimens. While it was easy to assure the desired Δ*t*_8/5_ = 10 s on the simulator, where the desired cooling curve is assured by the computer- controlled cooling system, for water quenching a special series of specimens had to be used to adjust the flow rate of water in the quenching bath.

Classic metallographic preparation was applied for examination of the microstructures, consisting of grinding, polishing, and etching. The average grain size was determined according to ASTM E112 [39] using the lineal intercept method. A Nikon Epiphot 300 optical microscope was used, equipped with an Olympus DP-12 digital camera.

The comparison of dislocation densities was omitted because:(a)Reduction of dislocation density due to recovery and recrystallization during the real welding and simulated weld thermal cycles are both diffusion-based processes. In previous research [18], it was confirmed that if specimens exhibit comparable austenitic grain sizes, comparable dislocation densities can be expected.(b)The cooling rates were equal in all cases; therefore, no significant differences of dislocation density could occur during cooling.

All the mechanical tests were performed at room temperature. For tensile tests and Charpy tests, 3 specimens of each type were used, and 1 specimen for fatigue crack growth.

Tensile tests were performed with a universal servo-hydraulic testing machine (Amsler 559/594), according to the standard EN ISO 6892-1, method B [40]. Standardized cylindrical specimens were used, with a diameter of 10 mm and a gauge length of 50 mm. Only material in as-received condition and AF + WQ specimens were tested, because WTCS specimens are unsuitable for standard tensile tests.

Vickers hardness HV 10 was measured with a Shimadzu HMV-2000 hardness tester in compliance with standard EN ISO 6507-1 [41]. ISO 9015–1 [42] was also considered for the real-weld FG HAZ. Prior to measurements, papers up to P 1200, according to ISO 6344-3 [43], were applied for grinding to assure smooth surfaces. Fifteen measurements were taken on each specimen, followed by calculation of the average values.

Instrumented impact tests were carried out with a Charpy pendulum Amsler RPK300 with a data-acquisition rate of 4 × 10^6^ readings per second according to standard ISO 148-1 [44]. Standard ISO-V-notch specimens were used for testing. Force versus time and energy versus time diagrams were recorded. Instrumented tests allowed the total energy for fracture *E*_t_ (*E*_t_ = *KV*_8_ according to ISO 148-1) to be split into the energy for crack initiation *E*_i_ and the energy for crack propagation *E*_p_.

For the fatigue crack growth tests, a standard method was applied according to ASTM E647-15e1 [45]. The geometry of the specimens is shown in Figure 3. A 1 mm-deep notch was prepared by wire electrical discharge machining. A strain gauge foil was attached on the side surface of the specimen, which enabled more precise crack length measuring during fatigue testing than required by the standard. The tests were carried out on a 160 Nm RUMUL Cractronic, and Fractomat equipment was used for fatigue crack growth measurement, Figure 4. All tests were performed with a bending load ratio *R* = 0.1.

## 3. Results and Discussion

### 3.1. Microstructures

Figure 5 shows the microstructures in the real-weld joint, from the weld metal (left) to the unaffected base metal (bottom right, around the scale bar).

The microstructures of the FG HAZs at higher magnifications are shown in Figure 6. The average grain sizes were 11.5 μm (real-weld FG HAZ), 12.6 μm (WTCS specimens), and 10.9 μm (AF + WQ specimens), and ASTM grain-size numbers *G* were 9.6, 9.33, and 9.75, respectively. The deviation from *G* of the real-weld FG HAZ was not higher than 0.27, whereby all microstructures can be regarded as quite similar, indicating that the applied *t*-*T* curves of both types of simulated samples were adequate.

### 3.2. Hardness Tests and Tensile Tests

The results of the hardness tests and tensile tests are presented in Table 4.

The average hardness of as-delivered (normalized) material was, as expected, significantly lower than the hardness of the FG HAZs, where the material was quench-hardened. The hardness of the AF + WQ samples, WTCS samples, and real-weld FG HAZ was very similar. Potentially, but because of the measured equal cooling rates not likely, the slightly higher hardness of the AF + WQ samples could be caused by different ways of cooling. The WTCS samples were clamped into water-cooled jaws, which resulted in a nearly one-dimensional heat flow from the hot central area of the specimen towards the jaws on both sides. The AF + WQ specimens were water-quenched, whereby all surfaces were cooled more intensively, and the heat flow from the center of the specimen was two-dimensional. Moreover, the average values differed by only about 1.5%, which is within the required repeatability range (deviations < 4% are required) according to ISO 6507-2 [41]. Therefore, the hardness of both types of simulated specimens can be regarded as practically equal. The real-weld FG HAZ exhibited slightly lower average hardness than the simulated specimens. However, the difference between the WTCS specimens and the real-weld FG HAZ was still within the required repeatability range of less than 4%. Therefore, with respect to hardness, the real-weld FG HAZ and both types of simulated specimens can be regarded as practically equal. Thereby, the hardness measurements, as did the comparison of microstructures, also confirmed that the simulated specimens were representative of a real-weld FG HAZ and that they were suitable for the intended further mechanical testing.

Standard specimens with a homogeneous microstructure along the whole gauge length are required for reliable tensile tests that can provide comparable results. Tensile tests of other conditions were omitted, as only material in as-delivered condition and AF + WQ allowed manufacturing of standard tensile test specimens exhibiting a homogeneous FG HAZ microstructure along the whole gauge length. If machined from a real-weld joint or from WTCS specimens, only a small fraction of the gauge length would exhibit the microstructure and mechanical properties of the FG HAZ, while the major fraction would exhibit different tensile properties, partially inferior to those of the FG HAZ. Consequently, the test pieces would not rupture in the FG HAZ area. Considering that in the real weld, the FG HAZ was only approximately 0.25 mm wide (Figure 5), even in the case of nonstandard U-groove test pieces, which enforce the fracture in the predetermined location, the results would be influenced strongly by the adjacent HAZ subzones. A further alternative could be micromechanical tests. However, with the decreasing cross section of test pieces, the influence of every local inhomogeneity like inclusions or other defects increases strongly, and the results obtained with microtensile tests can deviate significantly from those obtained with standard specimens.

Therefore, for all conditions, yield strength *R*_p02_ and tensile strength *R*_m_ were calculated from the average Vickers hardness according to Equations (1) and (2), respectively, proposed by Pavlina and Van Tyne [46]:*R*_p0.2_ = 2.876∙*HV* − 90.7 [MPa](1)
*R*_m_ = 3.734∙*HV* − 99.8 [MPa](2)

As Table 4 shows, the tested tensile properties and hardness in as-received condition deviated significantly from properties in other conditions. However, the tests in the as-received condition were not performed for comparison with other conditions, but to obtain more data for verification of Equations (1) and (2).

The measured and calculated *R*_p02_ and *R*_m_ values for the as-delivered material and for the AF + WQ specimens were in excellent agreement. In the AF + WQ specimens, the differences were even less than 1%. Consequently, since the WTCS specimens and the real-weld HAZ exhibited very similar microstructures and practically equal hardness to the AF + WQ specimens, it can be assumed that calculated *R*_p02_ and *R*_m_ of the WTCS specimens and real-weld FG HAZ are very close to real *R*_p02_ and *R*_m_ values.

### 3.3. Charpy Impact Tests

Charpy impact tests of the real-weld FG HAZ were also omitted for the same reasons as in the case of the tensile tests. The force vs. time (*F*-*t*) and energy vs. time (*E-t*) diagrams and fracture surfaces of the WTCS specimens and AF + WQ specimens are shown in Figure 7. The values are listed in Table 5. The total absorbed energies *E*_t_ were split into the energy for initiation of the crack *E*_i_ and the energy for crack propagation *E*_p_.

Compared to the AF + WQ specimens, the WTCS specimens absorbed notably higher total impact energy *E*_t_, higher energy for crack initiation *E*_i_, and higher energy for propagation *E*_p_. When evaluating the significance of the observed differences, the following points should be considered:(a)The maximum force measured was practically equal in both cases, slightly above 36 kN (Figure 7a,c), and the fracture surfaces showed similar characteristics of fracture (Figure 7b,d).(b)It should be mentioned that the differences in impact energies after rapid tempering and conventional tempering were already reported in the literature [30]. At the same HJP and practically identical hardness, the impact energies were higher after rapid tempering. Nevertheless, it must not be disregarded that in the research of Euser et al. [30], the HJP was equal in both cases, and most importantly some temperatures were in the range of temper embrittlement. Consequently, the differences in impact energies were caused by different extent of precipitation: during rapid tempering harmful precipitation was negligible, while during conventional tempering, it took place to its full extent. In our research, peak temperatures were always high enough to dissolve carbides, the Δ*t*_8-5_ was always the same, and not equality of HJPs, but comparable grain sizes was the priority.(c)Mechanical properties of metals depend on the real microstructure, where phase composition, phase fractions, dislocation density, and grain size are among the most important factors. Therefore, it seems admissible to assume that if the thermomechanical histories of different specimens assure comparable combinations of these factors, the impact toughness should also be similar.(d)The microstructures in the vicinity of the fracture surface also exerted influence on the results of the impact tests. While the entire volume of AF + WQ specimens exhibited homogeneous FG HAZ microstructure, the WTCS specimens exhibited the FG HAZ microstructure only in close proximity to the fracture surface, while the material in the adjacent IC and SC HAZ exhibited a lower yield point and higher ductility, which led to larger deformation of these areas. Consequently, the energy was not consumed only in the FG HAZ microstructure, but a considerable portion of it was consumed for plastic deformation of areas outside the FG HAZ. In this way, the test results for WTCS specimens were influenced by the existence of other HAZ subzones in the vicinity of the fracture area, and all the determined energies were too high. Consequently, it can be concluded that the different impact energies of the AF + WQ and WTCS specimens were not caused by different thermal cycles, but predominantly by the insufficient width of the simulated FG HAZ area in the WTCS specimens.

Finally, it should be pointed out that in all the tests, the initiation *E*_i_ was substantially higher than the propagation energy *E*_p_. This indicates that, regardless of the simulation method, at room temperature and in the absence of excessive amounts of hydrogen, the material in the FG HAZ was not particularly prone to crack occurrence. However, once initiated, cracks propagate quite easily with low energy consumption and are unlikely to stop.

### 3.4. Fatigue Crack Growth Tests

The results of the fatigue crack growth tests of the AF + WQ and WTCS specimens at ratio *R* = 0.1 are plotted in Figure 8. The fatigue crack growth thresholds for long crack propagation Δ*K*_thR_ were determined for both types of specimens. The crack growth rate d*a*/d*N* = 1 μm/10^6^ cycles was adopted as the long crack propagation threshold Δ*K*_thR_. The threshold Δ*K*_thR_ for the AF + WQ specimens was found to be 4.33 MPa m^0.5^, and 3.40 MPa m^0.5^ for the WTCS specimens. Determination of the coefficients C and m of the Paris curve were evaluated according to the Paris law, Equation (3), [47]. The results are summarized in Table 6.
d*a*/d*N* = C·Δ*K*^m^(3)

Comparison of the values revealed no significant differences. In both simulated FG HAZ microstructures, AF + WQ and WTCS, the coefficients C and m were very similar, which indicates similar crack propagation rates. The most obvious difference is that the WTCS specimen exhibited a lower threshold for long crack propagation Δ*K*_thR_ than the AF + WQ specimen. The seemingly big difference is most likely a consequence of the quite low value adopted as the threshold for long crack propagation. The ASTM E647 states the value d*a*/d*N* = 100 μm/10^6^ cycles as the most often adopted threshold for long crack propagation. However, crack length measurement with strain gauge foils is more precise than required by the standard, and therefore we could adopt a lower threshold (d*a*/d*N* = 1 μm/10^6^ cycles). Nevertheless, the lower threshold d*a*/d*N* means that by the time we start to consider the crack a microstructurally long one, the stress intensity factor *K* and the plastic zone around the crack tip were smaller, and consequently any microstructural barrier present in front of the crack tip can delay the propagation more efficiently. In the case of a higher adopted threshold d*a*/d*N*, the differences between the determined Δ*K*_thR_ values of both specimens would be smaller. Consequently, the difference of the determined thresholds Δ*K*_thR_ can be regarded as insignificant. Both the insignificant differences of Δ*K*_thR_, and the very similar values of *m*, indicated that the microstructures in both types of simulated specimens were very similar and can be regarded as equally suitable for fatigue crack growth tests.

More serious are the differences that can be observed when comparing the results for the FG HAZ with the results for the CG HAZ. In our previous research [23], the values Δ*K*_thR_ = 4.64 MPa m^0.5^, *C* = 4.55 × 10^−13^, and *m* = 4.17 were obtained for the CG HAZ. This means that under dynamic loads with low *R* ratios, the FG HAZ exhibits a slightly lower threshold Δ*K*_thR_ and is thereby more prone to long crack formation than the CG HAZ. However, individual HAZ subzones are quite narrow, and since CG HAZ does not exhibit a significantly higher Δ*K*_thR_, the crack will most likely not remain in the FG HAZ for very long, and once a long crack reaches the CG HAZ, its propagation rate will increase significantly.

## 4. Summary


(1)A single-V butt weld was made on a 15 mm-thick preheated 18CrNiMo7-6 plate to produce a reference microstructure, and Δ*t*_8/5_ = 10 s was measured during welding. The same value was adopted for the simulations of the FG HAZ microstructures.(2)The FG HAZ microstructures were prepared artificially (simulated) in two different ways: (a) using a weld thermal cycle simulator (WTCS) and (b) austenitizing in a laboratory furnace followed by water quenching (AF + WQ). The specimens were used for comparison of the microstructures for instrumented Charpy impact tests, hardness measurements, and fatigue crack growth tests at *R* = 0.1. The AF + WQ specimens were large enough for us to manufacture standard tensile test specimens.(3)The width of individual HAZ subzones in a real weld is very small. The results of mechanical tests of a real-weld HAZ, apart from the hardness measurement, cannot be linked to only one certain type of microstructure. Therefore, only the hardness of the real-weld FG HAZ was measured, while other mechanical tests were omitted on real-weld material. However, comparison of the hardness and microstructures indicated that other properties of the simulated specimens must also have been quite similar to those of the real-weld FG HAZ.(4)The microstructures of both types of simulated specimens were very similar to the microstructure of the real-weld FG HAZ. The ASTM grain-size numbers *G* were 9.6 (real-weld FG HAZ), 9.33 (WTCS), and 9.75 (AF + WQ specimens). The deviation from *G* of the real-weld FG HAZ of max. 0.27 confirmed that all three microstructures can be regarded as quite similar, thus indicating that the applied *t*-*T* curves of both types of simulated samples were adequate.(5)The hardness of all specimens was similar: AF + WQ specimens 425 HV, WTCS specimens 419 HV and the real-weld FG HAZ 405 HV. The differences were less than 4%, which is the required repeatability of measurements according to ISO 6507-2.(6)The absorbed impact energies of the WTCS specimens were higher (*KV* = 101.49 J) than the energies of the AF + WQ specimens (*KV* = 73.11 J). Typically, the width of simulated zones in WTCS specimens is quite small. Thereby, the softer and more ductile material outside the simulated zone also absorbs some of the impact energy, and thereby influences the results. On the contrary, in the AF + WQ specimens, all impact energy was absorbed in the simulated microstructure. Consequently, the results are more realistic, and the AF + WQ specimens can be regarded as more suitable for Charpy impact tests than WTCS specimens. Both types of specimens absorbed substantially more energy for crack initiation *E*_i_ than for crack propagation *E*_p_, thus indicating that the material is not very prone to crack initiation.(7)The results of the fatigue crack growth tests of both types of simulated specimens were very similar. The WTCS specimens exhibited Δ*K*_thR_ = 3.40 MPa m^0.5^, *C* = 1.64 × 10^−11^, and *m* = 2.4465. The AF + WQ specimens exhibited Δ*K*_thR_ = 4.33 MPa m^0.5^, *C* = 1.73 × 10^−11^, and *m* = 2.5114. The differences were insignificant and indicated very similar resistance to the occurrence of long cracks and very similar propagation rates. Thereby, the fatigue tests also confirmed the suitability of the AF + WQ specimens. Comparison of these results with the results of a previous investigation of a CG HAZ revealed the CG HAZ had a slightly higher threshold Δ*K*_thR_ than the FG HAZ, but once a long crack forms in a CG HAZ, it propagates significantly faster than in the FG HAZ.(8)Due to the narrow area of the simulated microstructure, WTCS specimens can be recommended without limitations only for investigation of microstructure, hardness measurements, and fatigue tests of FG HAZ. Comparison of the results for both types of simulated specimens between each other and with results that could be obtained reliably for a real-weld HAZ confirmed that the AF + WQ specimens are suitable for mechanical tests of simulated FG HAZs. An important advantage of the AF + WQ specimens over WTCS specimens was that the whole volume of the AF + WQ specimens exhibited homogeneous simulated microstructure and thereby enabled standard tensile tests (impossible with WTCS specimens) and more accurate Charpy impact testing.


## 5. Future Work

Comparison of crack propagation thresholds in FG HAZs and CG HAZs revealed that the threshold is lower in the FG HAZ. Therefore, in the future, the focus will be on FG HAZs; more specifically, on initiation and early propagation of cracks from defects. The defects in simulated FG HAZ microstructure will be made artificially by indentation and hole drilling. As the residual stress due to the creation of defects affects crack initiation and early crack propagation, the influence of residual stress will be studied in detail.

## Figures and Tables

**Figure 1 materials-15-06782-f001:**
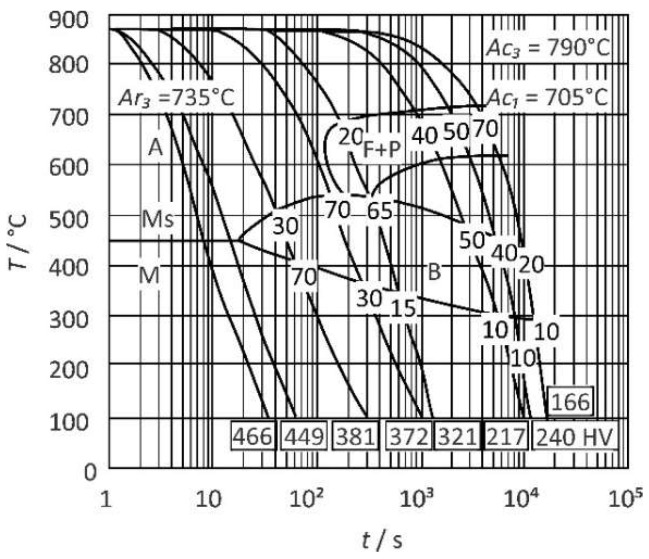
CCT diagram for 18CrNiMo7-6 steel. Redrawn from the material data sheet [33].

**Figure 2 materials-15-06782-f002:**
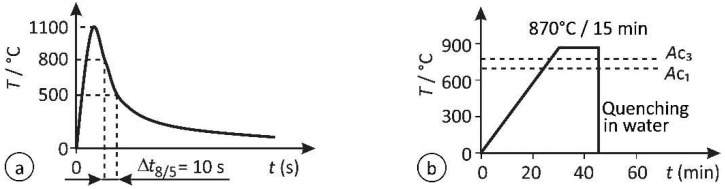
Thermal cycles of the simulated specimens: (**a**) weld thermal cycle simulator; (**b**) austenitizing in a laboratory furnace + water quenching.

**Figure 3 materials-15-06782-f003:**
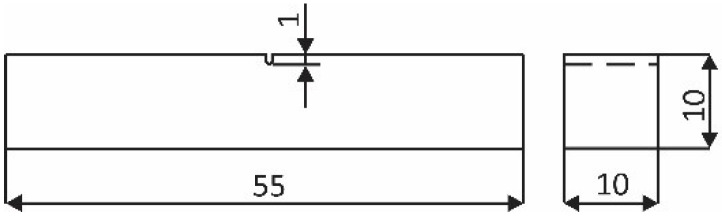
Geometry of the specimen for fatigue growth tests according to Standard ASTM E647-15e1 [45].

**Figure 4 materials-15-06782-f004:**
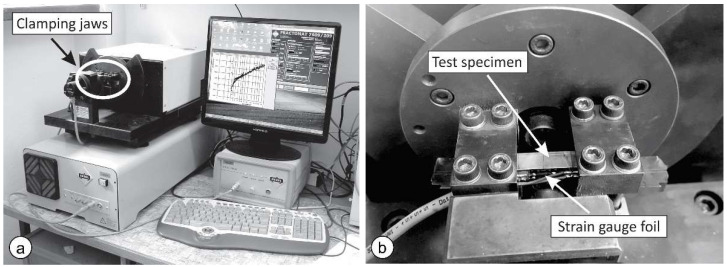
(**a**) The RUMUL Cracktronic and Fractomat machine; (**b**) specimen in the clamping jaws.

**Figure 5 materials-15-06782-f005:**
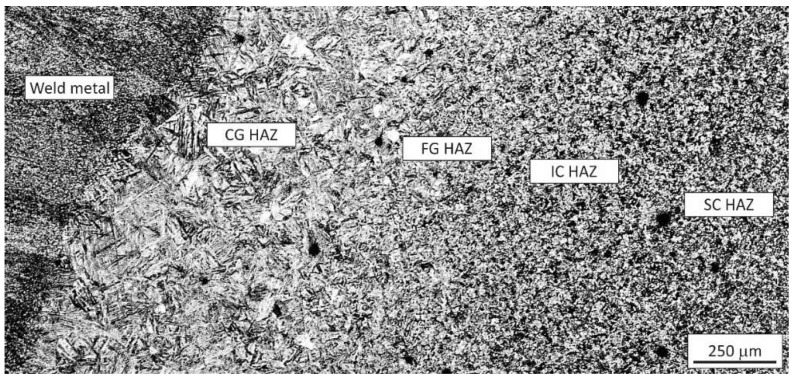
Microstructures in the real-weld joint (from left to right): weld metal, CG HAZ, FG HAZ, IC HAZ, and SC HAZ.

**Figure 6 materials-15-06782-f006:**
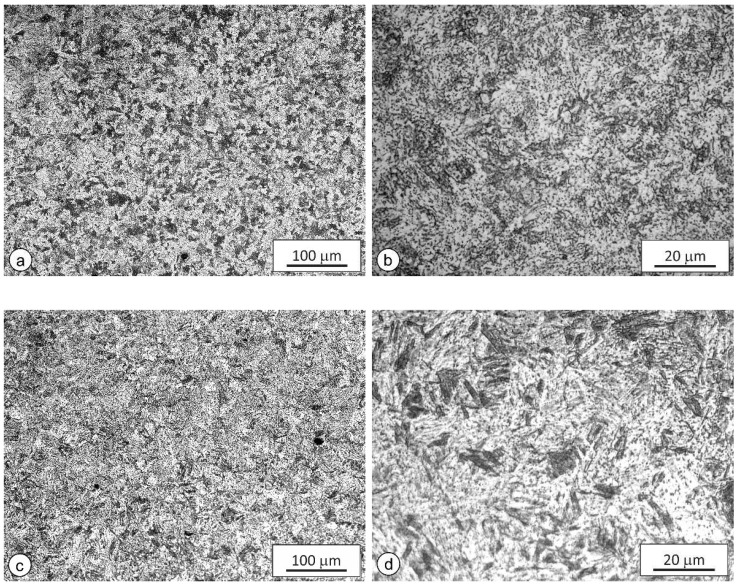
Microstructure of FG HAZs: (**a**) real-weld FG HAZ, 200×; (**b**) real-weld FG HAZ, 1000×; (**c**) WTCS specimen, 200×; (**d**) WTCS specimen, 1000×; (**e**) AF + WQ specimen, 200×; (**f**) AF + WQ specimen, 1000×.

**Figure 7 materials-15-06782-f007:**
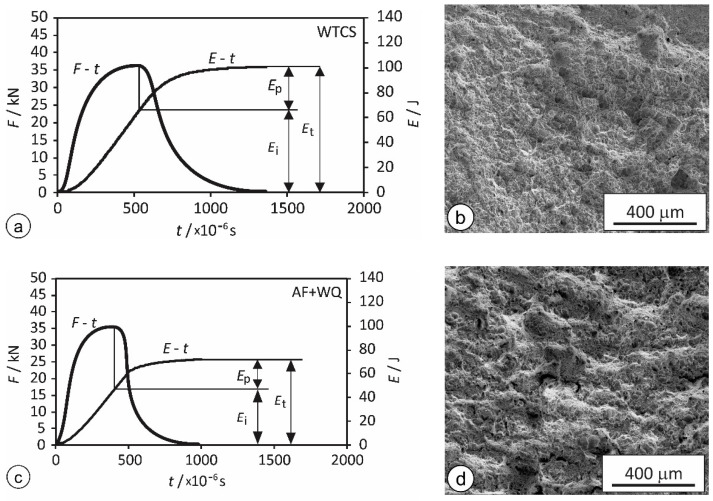
*F-t* and *E-t* diagrams of the instrumented Charpy tests and SE-SEM images of the fracture surfaces: (**a**,**b**) WTCS specimens; (**c**,**d**): AF + WQ specimens.

**Figure 8 materials-15-06782-f008:**
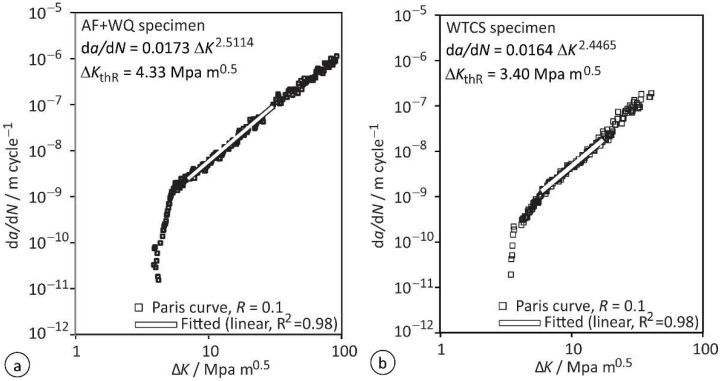
Results of the fatigue crack growth test: (**a**) AF + WQ specimen; (**b**) WTCS specimen.

**Table 1 materials-15-06782-t001:** Chemical composition of the 18CrNiMo7-6 steel (wt.%).

Chemical Element	C	Si	Mn	P	S	Cr	Ni	Cu	Mo	Al
wt. %	0.18	0.22	0.43	0.012	0.028	1.56	1.48	0.15	0.28	0.023

**Table 2 materials-15-06782-t002:** Welding parameters for real welding.

*I*/A	105
*U*/V	24.2
*v*_welding_/cm min^−1^	14.6
*Q*/kJ cm^−1^	8.9
Coated electrode EN ISO 18275-A [38]	E 89 6 ZB62 H5, ϕ3.2 mm
*T*_preheat_/°C	200
*T*_interpass_/°C	250
Δ*t*_8/5_/s	10

**Table 3 materials-15-06782-t003:** Thermal cycles of the simulated specimens.

	WTCS	AF + WQ
*T* _preheat_	200 °C	-
Heating rate	150 °C s^−1^	Approx. 2.8 °C s^−1^ *
*T* _peak_	1100 °C	870 °C
*t* _hold_	0.5 s	10 min *
Δ*t*_8/5_	10 s	10 s

* Center of cross section.

**Table 4 materials-15-06782-t004:** Results of tensile tests and hardness tests.

Specimen		*R*_p02_/MPa	*R*_m_/MPa	*A*/%	HV 10/−
As delivered	Tensile test Calculated *	484 464	634 621	26	193
AF + WQ	Tensile test Calculated *	1121 1132	1475 1487	9.05	425
WTCS	Calculated *	1114	1465	-	419
Real-weld FG HAZ	Calculated *	1074	1412	-	405

* Calculated according to Equations (1) and (2).

**Table 5 materials-15-06782-t005:** Results of the instrumented Charpy tests for both types of simulated FG HAZ specimens.

	*E*_t_ = *KV*_8_/J	*E*_i_/J	*E*_p_/J	Portion of Ductile Fracture/%
WTCS	101.49	58.94	42.55	51.89
AF + WQ	73.11	49.7	23.41	55.41

**Table 6 materials-15-06782-t006:** Results of the fatigue crack growth tests.

	*R*/−	Δ*K*_thR_/MPa m^0.5^	*C*	*m*
AF + WQ	0.1	4.33	1.73 × 10^−11^	2.5114
WTCS	0.1	3.4	1.64 × 10^−11^	2.4465

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
