# Peer review of "Simulation and Mechanical Properties of Fine-Grained Heat-Affected Zone Microstructure in 18CrNiMo7-6 Steel"

_materials, 2022, doi:10.3390/ma15196782_

Round 1
Reviewer 1 Report
Authors have made an excellent effort to simulate microstructure of the most sensitive part of weld joint, in order to establish mechanical properties of this critical zone. In order to improve this paper, I have following suggestions:
1. INTRODUCTION is very long/extensive with a lot of non-necessary details:
a. detailed evaluation of references (not referring, but explaining each of them)
b. details about non-QT steels
c. details on common knowledge and basics in welding
Introduction is written from row19 to row 158 (almost three pages). I suggest to shorten text and give only necessary information.
2. MATERIALS AND METHODS Authors again give too many details. For each standpoint, they ellaborate original paper/reference, instead of only
a. Again, detailed evaluation of references (not referring, but explaining each of them). Chapter MATERIALS AND METHODS is written from row160 to row 272 (almost three and half pages). I suggest to shorten text and give only necessary information.
b. When speaking of steel, rows 164-166 repeat general knowledge on Q+T steels. It should be deleted, or rephrased in Introduction.
c. row 222 – “As grain growth and dislocation density during heating are both diffusion based processes”, is not accurate to point out, since dislocations can be generated ONLY by shear stresses. On the other hand, with heating of deformed metal, recovery and recristallization occurs. These two processes ARE diffusion based processes. Please correct this in text.
3. RESULTS AND DISCUSSION.
a. do not use term “photographed” for microstructures. Also, if a magnification bar is given on microphotograph,it is not necessary to write maagnification. For example:
Instead of Figure 9. Microstructure of FG HAZs photographed at magnifications 200× and 1000×; a) and b): Real weld FG HAZ; c) and d): WTCS specimen; e) and f): AF + WQ specimen.
Should be Figure 9. Microstructure of FG HAZs: (a) Real weld FG HAZ, 200x; (b) Real weld FG HAZ, 1000x (c) WTCS specimen, 200x; (d) WTCS specimen, 1000×; (e) AF + WQ specimen 200x; (f) AF + WQ specimen 1000×.
b. Row 282 – “The average grain size vas determined according to ASTM E112 [47] using the lineal intercept method”. – This text should be in experimental part
c. Row 379 – whole paragraph “a)” is irelevant and should be deleted.
d. Row 395 – When mentioning EUSER et al , please ADD reference number.
e. Rows 404 – 408 – This generalization can not take toughness into account.
f. Row 436 – close values of m indicates that microstructure is very simillar
4. CONCLUSIONS. As listed in paper, this chapter should be renamed as SUMMARY. Conclusion 8 is too general statement, should be deleted.
It is my opinion that shorter version of this paper will be much better.
Reviewer 2 Report
1. Abstract should be given as more interesting. Express at least one of the main aspects and features of the paper.
2. Wherever applicable, the scientific explanation needs to be added and the research novelties need to be clearly emphasized.
3. The gap area in the research is not clear.
4. At the end of Introduction section, it would be better to add the paper's organization in different sections.
5. There is no scientific justification for the selection of welding parameters considered.
6. Further, results and analysis of experiments should be compared with previous researchers by citing references
7. Scope for Future work is missing.
8. Please check the manuscript for wrong choice of words, grammatical errors and incoherent sentence structure.
Reviewer 3 Report
This paper describes simulation and mechanical properties of fine grain heat affected zone microstructure in 18CrNiMo7-6 steel. To investigate the mechanical properties of the fine-grain heat-affected zone in the 18CrNiMo7-6 steel, two different methods for simulation of as-welded microstructures were used. a) a weld thermal cycle simulator, and b) austenitizing in a laboratory furnace + water quenching. Mechanical properties were investigated, and microstructures were compared. Comparison of results obtained with simulated specimens and a real weld FG HAZ confirmed that both types of simulated specimens are suitable for mechanical tests of individual HAZ subzones. Due to the use of simulated test specimens, the determined mechanical properties can be linked to the FG HAZ microstructure in 18CrNiMo7-6 steel. This paper is interesting, well written to a large extent and well fits the scope of the journal. However, small revisions are necessary before publication.
[1] Title
“8CrNiMo7-6 steel” must be changed to “18CrNiMo7-6 steel”.
[2] Abstract
“To investigate the mechanical properties of a the fine-grain heat-affected zone” must be changed to “To investigate the mechanical properties of the fine-grain heat-affected zone”.
[3] Introduction
Abbreviation should be defined like TRIP (Transformation Induced Plasticity).
[4] Results and discussion
The explanation of figure 8 is mission. Please add it before the figure.
[5] Line 90 The authors describe that “Much less data is available on FG HAZ in 18CrNiMo7-6 steel.” Please cite some examples to compare the result in the present paper.
[6] Line 232 The authors described that “Only material in as-received condition and WTCS specimens were tested.”, but the latter should be AF+WQ.
[7] Line 283 The authors described that “In all three cases the average grain size was about 10 m”. Please give the value in each case. The size in real weld FG HAZ seems to be different from other two cases.
[8] Line 502 This conclusion is not described in the results and discussion. It is necessary to describe it also in the results and discussion.
Round 2
Reviewer 2 Report
Authors have made the significant changes in the revised manuscript. Hence, accept the manuscript for publication in its present form.